Oligotrophic wetland sediments susceptible to shifts in microbiomes and mercury cycling with dissolved organic matter addition

Graham Emily B. emily.graham@pnnl.gov 1
Gabor Rachel S. 2
Schooler Shon 3
McKnight Diane M. 4 5 6
Nemergut Diana R. 4 7
Knelman Joseph E. 4
1 Biological Sciences Division, Pacific Northwest National Laboratory , Richland , WA , United States of America
2 School of Environment and Natural Resources, Ohio State University , Columbus , OH , United States of America
3 Lake Superior National Estuarine Research Reserve , Superior , WI , United States of America
4 Institute for Arctic and Alpine Research, University of Colorado at Boulder , Boulder , CO , United States of America
5 Civil Engineering Department, University of Colorado at Boulder , Boulder , CO , United States of America
6 Environmental Studies Program, University of Colorado at Boulder , Boulder , CO , United States of America
7 Biology Department, Duke University , Durham , NC , United States of America
Smoak Joseph
Electronic publication date: 2018 Apr 3
Publication date: 2018
Volume: 6
Electronic Location ID: e4575
Received 2018 Feb 1; Accepted 2018 Mar 15
Copyright: ©2018 Graham et al.
Copyright year: 2018
Copyright holder: Graham et al.
License: This is an open access article distributed under the terms of the Creative Commons Attribution License, which permits unrestricted use, distribution, reproduction and adaptation in any medium and for any purpose provided that it is properly attributed. For attribution, the original author(s), title, publication source (PeerJ) and either DOI or URL of the article must be cited.
License URL: https://creativecommons.org/licenses/by/4.0/

Keywords: Mercury methylation, Fermentation, Carbon, Microbial community structure, Sediment microbiome, Mercury contamination, Freshwater microbiology, Wild rice, Fluorescence spectroscopy, Organic matter chemistry

Funding: EPA STAR NOAA NERRS US Department of Energy (DOE) Office of Biological and Environmental Research (BER) Subsurface Biogeochemical Research Program’s Scientific Focus Area (SFA) Pacific Northwest National Laboratory (PNNL) This work was supported by EPA STAR and NOAA NERRS fellowships to Emily B. Graham and a JGI CSP grant to Diana R. Nemergut. The authors received support from the US Department of Energy (DOE), Office of Biological and Environmental Research (BER), as part of Subsurface Biogeochemical Research Program’s Scientific Focus Area (SFA) at the Pacific Northwest National Laboratory (PNNL). PNNL is operated for DOE by Battelle under contract DE-AC06-76RLO 1830. The funders had no role in study design, data collection and analysis, decision to publish, or preparation of the manuscript.

==============================
Recent advances have allowed for greater investigation into microbial regulation of mercury toxicity in the environment. In wetlands in particular, dissolved organic matter (DOM) may influence methylmercury (MeHg) production both through chemical interactions and through substrate effects on microbiomes. We conducted microcosm experiments in two disparate wetland environments (oligotrophic unvegetated and high-C vegetated sediments) to examine the impacts of plant leachate and inorganic mercury loadings (20 mg/L HgCl2) on microbiomes and MeHg production in the St. Louis River Estuary. Our research reveals the greater relative capacity for mercury methylation in vegetated over unvegetated sediments. Further, our work shows how mercury cycling in oligotrophic unvegetated sediments may be susceptible to DOM inputs in the St. Louis River Estuary: unvegetated microcosms receiving leachate produced substantially more MeHg than unamended microcosms. We also demonstrate (1) changes in microbiome structure towards Clostridia, (2) metagenomic shifts toward fermentation, and (3) degradation of complex DOM; all of which coincide with elevated net MeHg production in unvegetated microcosms receiving leachate. Together, our work shows the influence of wetland vegetation in controlling MeHg production in the Great Lakes region and provides evidence that this may be due to both enhanced microbial activity as well as differences in microbiome composition.

Introduction

Mercury methylation in anoxic sediments is central to the bioaccumulation of mercury in plant and animal tissue (Benoit et al., 2003; Morel, Kraepiel & Amyot, 1998; Ullrich, Tanton & Abdrashitova, 2001) and poses a significant environmental and human health concern in the freshwater wetlands of the Great Lakes region (Branfireun et al., 1999; Harmon et al., 2005; Jeremiason et al., 2006). Dissolved organic matter (DOM) has been a focus of geochemical investigations for decades, and both positive and negative interactions between DOM and mercury methylation—principally, a microbial transformation (Hsu-Kim et al., 2013; Paranjape & Hall, 2017)—have been demonstrated under contrasting environmental conditions (Graham, Aiken & Gilmour, 2013; Hsu-Kim et al., 2013; Ravichandran, 2004). Further, recent discoveries in microbial ecology of mercury methylation have highlighted the complex roles of diverse microbial communities in driving mercury cycling. Yet linkages among DOM cycling, sediment microbiomes that directly mediate mercury methylation, and MeHg production remain poorly described.

Dissolved organic matter is comprised of various classes of organic compounds (primarily organic acids) with a wide range of molecular weights and aromaticities (Lambertsson & Nilsson, 2006; Wetzel, 1992). DOM concentrations are elevated in wetlands relative to other freshwater systems (>10 mg/L), and the humic fraction derived from plant leachate predominates (Fellman, Hood & Spencer, 2010). With respect to mercury cycling in wetlands, mercury methylation is impacted both by binding properties of the humic DOM fraction, resulting either in increased dissolution of inorganic mercury complexes or in physical inhibition of mercury bioavailability (Drexel et al., 2002; Haitzer, Aiken & Ryan, 2002; Waples et al., 2005), and by the provisioning of organic substrate for microbial activity (Hsu-Kim et al., 2013; King et al., 2000; Lambertsson & Nilsson, 2006). Since mercury methylation is strongly impacted by DOM, environments such as the Great Lakes’ St. Louis River estuary, which contains areas of both vegetated and unvegetated sediments, may also show differences in the capacity for MeHg production across vegetation gradients that exhibit pronounced differences in DOM content.

Beyond the interaction of DOM and mercury and the influence of DOM on microbial activity, numerous studies have shown that microbiome composition itself is influenced by DOM quantity and/or quality (Docherty et al., 2006; Forsström, Roiha & Rautio, 2013; Graham et al., 2016a; Graham et al., 2017; Pernthaler, 2013; Stegen et al., 2016), and such changes in environmental microbiomes alter biogeochemistry (Graham et al., 2016a; Graham et al., 2017; Graham et al., 2016b). The composition of microbial communities has recently gained increased attention with regard to mercury cycling due to the discovery of the hgcAB gene cluster, which has allowed investigations into the microbial ecology of mercury cycling (Boyd et al., 2017; Gilmour et al., 2013; Gionfriddo et al., 2016; Paranjape & Hall, 2017; Parks et al., 2013; Rani et al., 2015; Rothenberg et al., 2016; Schwartz et al., 2016). Such work has increased knowledge on the microbiology of mercury methylation, expanding potential microorganisms mediating methylation beyond sulfate-reducing bacteria (Compeau & Bartha, 1985; Hsu-Kim et al., 2013), iron-reducing bacteria (Kerin et al., 2006) and methanogens (Hamelin et al., 2011). For example, Gilmour et al. (2013) have identified five clades of putative methylators, including new clades of syntrophic and Clostridial organisms.

In this study, we examine the influence of DOM from plant leachate on net methylmercury (MeHg) production in a contaminated freshwater estuary at the base of Lake Superior. First, we describe net MeHg production in environments exhibiting high (vegetated sediments) and low (unvegetated sediments) DOM concentration. These two contrasting sediment types are widely found in the environment. We hypothesize that both DOM quantity and quality influence mercury methylation through two different pathways, (1) by regulating microbial activity and (2) by shifting bacterial community composition, and therein the metabolic diversity of mercury methylators. We test this hypothesis across chemically distinct sediments associated with unvegetated (oligotrophic, low-C) and vegetated (high-C) environments. We used a microcosm experiment to monitor changes in sediment microbiomes, DOM chemical quality, and net MeHg production in response to additions of leachate from overlying plant material and to high levels of inorganic mercury. In total, this work delineates a broad view of how vegetated vs. unvegetated sediments in the Great Lakes’ St. Louis River Estuary may have different capacities for the cycling of mercury.

Methods

Field site

The St. Louis River Estuary is home to the largest US port on the Great Lakes and covers roughly 12,000 acres of wetland habitat directly emptying into Lake Superior. Mining in the headwaters, industrial discharge in the port, and atmospheric deposition have left a legacy of mercury contamination in the sediment. We obtained sediment samples from nearshore vegetated (Zizania palustris (wild rice), 46°40.855′N, 91°59.048′W) and unvegetated (46°41.918′N, 92°0.123′W) patches in Allouez Bay and fresh wild rice plant matter from nearby Pokegama Bay (46.683448°N, 92.159261°W). The purpose of this design was to obtain sediment from locations in close proximity to each other and to minimize our ecological impact on sensitive wild rice patches by gathering plant material from a larger nearby vegetation patch. The overlying water column was <1 m at each location. Both habitats are clay-influenced embayments that drain an alluvial clay plain created by deposition during the retreat of the last glaciation approximately 10,000 years BP.

Experimental design

A total of 20 anoxic microcosms were constructed in September 2013 to investigate relationships between DOM cycling, sediment microbiomes, and mercury methylation. Microcosms were constructed in 500-mL airtight glass mason jars and stored at room temperature in the dark in Mylar bags with oxygen-absorbing packets between subsampling to maintain anoxia. Sediment was obtained in 250-mL amber Nalgene bottles from the top 10 cm of sediment (grab samples) using a block sampling design described in Supplemental Information 1. Leachate was extracted using 1 g dried, ground plant matter:20 mL of Nanopure water, shaken for 1 hr, rested for 4 hr, and filtered through Whatman 0.7 µm GFF filters (Whatman Incorporated, Florham Park, NJ, USA). All materials (except filters) were acid washed, and filters were combusted to remove residual C. The chemical composition of leachate is available in Supplemental Information 1. Our experiment was designed to promote microbial MeHg production (1) by adding an abundance of inorganic mercury. (2) taking reasonable measures to minimize abiotic photo-methylation and -demethylation (Morel, Kraepiel & Amyot, 1998), and (3) sustaining a low redox environment for inhibiting demethylation (Compeau & Bartha, 1984). We acknowledge that we did not eliminate all potential demethylation activity from our microcosms, and we refer to changes in MeHg as ‘net MeHg production’ to reflect possible demethylation (consistent with other recent work, Schwartz et al., 2016). All experimental set up and sample processing was conducted in an anaerobic glovebox containing 85% N2, 5% CO2, and 10% H2 gas mix at the USGS in Boulder, CO. Jars were degassed in the glovebox for 48 hr prior to experimentation to remove oxygen.

A full-factorial design was employed with two environments (vegetated and unvegetated sediment) and two treatments (plant leachate and Nanopure water). Sediments were homogenized via mixing but unsieved to maintain environment characteristics. Large roots (>1 cm) were infrequent and removed to lessen heterogeneity among replicates. Each microcosm received 100 g wet sediment and 250 mL solution consisting either of leachate at 100 mg/L (∼5× natural concentrations to mimic a loading event) and HgCl2 at 20 mg/L (50 µg/g wet sediment) in Nanopure water (leachate replicates) or solely of HgCl2 at 20 mg/L in nanopure water (no leachate replicates). The purpose of HgCl2 addition at high concentration was to counteract initial differences in mercury, minimize HgCl2 inaccessibility due to abiotic organo-metal interactions, and provide substrate for the duration of the experiment. HgCl2 concentrations were elevated to extreme levels (1,000× ambient concentration), reflecting a convention in ecological change literature to stress ecosystems and eliminate substrate limitation with high levels of nutrients to assist in deciphering mechanistic controls over a process of interest. Because we were interested in the ability of microorganism to generate methylmercury when given an unlimited supply of inorganic mercury, we did not parse the origin of MeHg (i.e., from mercury originally in the sediment vs. from added substrate). Although we added more HgCl2 than is common in mercury-DOM literature, we note that (1) the duration of experiment was long relative to other studies (28 days vs. <24 hr in many studies); (2) DOM concentrations are high in the St. Louis River Estuary (>20 mg/L in the water column), and (3) added HgCl2 concentrations were of comparable magnitude to some microcosm experiments of similar design (Harris-Hellal et al., 2009; Ruggiero et al., 2011; Zhou et al., 2012). Though we did not directly assess microbial activity, we estimate minimal dosage effects as communities without leachate did not change through time in unvegetated microcosms and only slightly changed through time in vegetated microcosms (R2 = 0.19, see results and Fig. S1). Microcosms were incubated for 28 days, and subsamples of sediment and water were taken every seven days for analysis of sediment microbiomes and DOM characteristics.

Sediment chemistry and mercury methylation

Percent carbon and nitrogen, NO3−/NO2−, NH4+, total particulate organic carbon (TPOC), total dissolved nitrogen (TDN), and pH were determined on pre-incubation sediments, as described in Supplemental Information 1. For total- and methylmercury analysis, initial (day 0) and final (day 28) subsamples were frozen at −70 °C, freeze-dried, and sent on dry ice to the USGS Mercury Lab in Middleton, WI for analysis by aqueous phase ethylation, followed by gas chromatographic separation with cold vapor atomic fluorescence detection (Method 5A-8), acid digestion (Method 5A-7), and QA/QC. Mercury analyses were performed on three of five replicates for each environment and microcosm type. All other analyses were performed on five replicates, except for no unvegetated microcosms without leachate beyond day 0 (n = 4, one replicate destroyed during experiment).

Dissolved organic matter characteristics

Aqueous subsamples from water overlying sediments were collected at 7-day intervals (days 0, 7, 14, 21, and 28) to determine non-purgeable organic carbon (NPOC) concentration and specific UV absorbance at 254 nm (SUVA254) as well characteristics of the optically active DOM pool (mostly associated with humic DOM fraction), as described in Supplemental Information 1. We calculated the fluorescence index (FI) to determine the relative contribution of microbial vs. terrestrial matter to the DOM pool, the humic index (HIX) to identify large aromatic compounds consistent with humic material, and the freshness index to determine the availability of labile carbon (Fellman, Hood & Spencer, 2010; Gabor et al., 2014a) using MATLAB software (2013a; The MathWorks, Natick, MA, USA) according to Gabor et al. (2014b).

Microbial DNA extraction, 16S rRNA amplicon, and metagenomic shotgun sequencing

DNA from each sediment subsample was extracted using the MO Bio Power Soil DNA Extraction kit (MO BIO Laboratories, Carlsbad, CA, USA), as described in Knelman et al. (2017), Knelman et al. (2015) and Castle et al. (2016). The region encoding the V4 fragment of the 16S rRNA gene was amplified with the primers 515F/806R, using the PCR protocol described by the Earth Microbiome Project (Caporaso et al., 2012) (Supplemental Information 1). The final multiplexed DNA samples were sequenced at CU-Boulder (BioFrontiers Institute, Boulder, CO) on an Illumina MiSeq with the MiSeq Reagent Kit v2, 300 cycles (Illumina, Cat. # MS-102-2002) to generate 2 × 150-bp paired-end reads. Sequences are available at https://figshare.com/articles/Sequences/5833935. In addition, 3 unvegetated leachate replicates at day 0 (before leachate addition) and day 28 were sent to the Joint Genome Institute (JGI) for shotgun metagenomic sequencing on the Illumina HiSeq platform. Sequences are available at img.jgi.doe.gov under GOLD Study ID ‘Gs0113736’.

To examine shifts in bacterial community composition that may relate to mercury cycling, we located 90 of 142 (65%) microbial strains that have been identified as containing the hgcAB gene cluster (listed by Oak Ridge National Laboratory (ORNL), http://www.esd.ornl.gov/programs/rsfa/data/PredictedMethylators/PredictedMethylators_20160420.pdf) with available complete or partial 16S rRNA gene sequences in the NCBI GenBank database. While our database was not exhaustive and sub-OTU level sequence variation (97%) may impact an organism’s methylation potential, the purpose of this analysis was to identify possible methylating OTUs of interest, and the represented sequences spanned all clades of methylators (Fig. S2). We created a closed-reference database of these sequences and picked OTUs against this database in QIIME to discern a reduced set of potential methylating taxa present in our samples. This highlighted taxa that are known to contain methylating organisms. We assessed how such taxa shift in response to DOM addition across the two sediment types and may correspond with changes in mercury methylation.

Sequence analysis

Partial 16S rRNA genes were filtered for sequence length and minimum quality score in the UPARSE pipeline (Edgar, 2013) and OTUs were assigned using QIIME (Caporaso et al., 2010) (Supplemental Information 1). Metagenomic shotgun sequences were assembled and classified against the protein families database (Pfam) (Finn et al., 2013), Clusters of Orthologous Groups of proteins (COG) (Tatusov et al., 2003), and Kyoto Encyclopedia of Genes and Genomes (KEGG) (Kanehisa & Goto, 2000) by JGI via the IMG database pipeline (Markowitz et al., 2012). In addition, a BLAST database was constructed from all hgcA and hgcB gene sequences available in GenBank. A BLASTX search was conducted against this database to identify taxonomic affiliation of methylators in our samples; however, our query resulted in no matches, likely due to inadequate sequencing depth.

Statistical analysis

All analyses, unless otherwise noted, were conducted using the R software platform. Shapiro–Wilk tests were used to verify normality and assess the appropriateness of parametric vs. non-parametric tests. Multivariate sediment properties (e.g., sediment geochemistry) were compared across environments at day 0 with Hotelling’s T-square Test and post hoc Student’s t-tests. MeHg production was calculated by subtracting day 0 from day 28 MeHg concentrations; values below detection limit were assigned the detection limit as a value for a conservative estimate of change. MeHg production was compared across groups using ANOVA. Changes in DOM indices (FI, freshness, HIX) through time (days 0, 7, 14, 21, and 28) in each sample group were assessed with linear and quadratic regressions. DOM samples with SUVA254 >7 were removed due to fluorescence interference from inorganic molecules. Comparisons of DOM indices between data subsets were conducted with ANOVA and post hoc Tukey HSD.

Microbial community dissimilarity matrices based on 16S rRNA sequences were constructed using the weighted UniFrac method in QIIME (Lozupone et al., 2011). Alpha diversity for each sample was assessed using the PD whole tree metric in QIIME. Changes in community structure through time (days 0, 7, 14, 21, 28) were assessed with ANOSIM in QIIME. Differences in alpha diversity at day 0 were assessed using unpaired one-way Student’s t-tests. Relative abundances of major clades were assessed between vegetated and unvegetated environments at day 0 and changes in clades through time (days 0, 7, 14, 21, 28) were assessed using non-parametric Kruskal–Wallis tests with FDR-correct P values. SIMPER analysis was conducted using the ‘vegan package’ to identify OTUs associated with community dissimilarity between days 0 and 28 in microcosms receiving leachate.

To focus our analysis of microbiome composition on taxa that may contain methylating bacteria, we used one-sided Mann–Whitney U tests to identify clades of organisms on ORNL’s list of potential methylators that changed through time. We build upon previous work by Rothenberg et al. (2016), who examined genus-level changes in known methylating clades, and Rani et al. (2015), who examined only methylating Deltaproteobacteria at the OTU level, by targeting all microorganisms identified by ORNL at the OTU level. We present results as changes in percent relative abundance rather than fold-change due to the absence of some organisms at day 0 (i.e., zero abundance). Microorganisms that exhibited significant changes were further compared to HIX and net MeHg production to examine potential relationships among DOM, bacterial taxa, and mercury cycling. For this analysis, we used the Pearson product-momentum correlation coefficient, grouping leachate and no leachate microcosms within each environment in a single analysis to provide replication across a wide range of variation (analysis conducted on data from day 28, n = 6).

Finally, we explored metagenomic shotgun sequences for information on specific microbial metabolic pathways that changed through time in our microcosms. We used binomial tests to detect increases in the frequency of COGs, Pfams, and KEGG pathways at day 28 relative to day 0. Targets more abundant at day 28 (FDR-corrected P < 0.01) were examined for correlations with HIX and MeHg production with the Pearson product-momentum correlation coefficient to decipher possible links between microbial metabolism, DOM cycling, and net MeHg production.

Results

Ambient geochemistry and microbiology

Physicochemical properties and sediment microbiomes differed between vegetated and unvegetated environments (Hotelling P = 0.004, Table 1). The unvegetated sediment was extremely oligotrophic compared to vegetated sediment, with much lower concentrations of C and N, and both vegetated and unvegetated environments appeared to be N-limited (C:N 16.43 and 20.06). In addition, net MeHg production in sediments without leachate addition was significantly higher in vegetated sediment than unvegetated sediment, by nearly two orders of magnitude (Fig. 1). Final total Hg concentrations at the end of the microcosm experiment were 79.2 ± 18.3 (vegetated with leachate), 86.5 ± 14.5 (vegetated without leachate), 15.0 ± 5.8 (unvegetated with leachate), and 18.1 ± 1.3 (unvegetated without leachate) µg/g wet sediment. Initial concentrations are listed in Table 1. Microbial community structure and alpha diversity were significantly different between the two environments (ANOSIM, P = 0.001, R = 1.00, t-test, P = 0.01), though major phyla were similar (Table 1).

Table 1 Mean chemical and biological characteristics of vegetated (n= 5) and unvegetated (n = 5) environments are presented Table 1.

All data are derived from sediments. Asterisks represent significant differences from post hoc t-tests, and standard deviations are presented in parentheses. Microbial groups are listed as relative abundance (fraction of all OTUs in community).

	Vegetated environment	Unvegetated Environment	
pH*	5.6 (0.09)	5.8 (0.40)	
Water content (g dry/g wet)***	0.54 (0.04)	0.85 (0.04)	
NH4 (ug/g wet)***	5.96 (1.28)	1.44 (0.56)	
TPOC (ug/g wet)***	90.4 (4.8)	7.2 (4)	
TDN (ug/g wet)**	4.8 (1.6)	3.2 (0.8)	
percent C***	13.16 (2.20)	1.82 (3.39)	
percent N***	0.8 (0.06)	0.1 (0.23)	
C:N*	16.43 (1.59)	20.06 (5.36)	
MeHg (ng/g)**	2.67 (2.18)	0.24 (0.12)	
THg (ng/g)	306.56 (551.07)	3.16 (3.99)	
MeHg:THg	0.02 (0.009)	0.32 (0.45)	
Proteobacteria***	0.3 (0.04)	0.43 (0.02)	
Chloroflexi***	0.17 (0.01)	0.06 (0.009)	
Bacteroidetes	0.11 (0.02)	0.13 (0.03)	
Acidobacteria*	0.07 (0.009)	0.08 (0.02)	
Nitrospirae***	0.05 (0.009)	0.02 (0.009)	
Actinobacteria***	0.03 (0.007)	0.07 (0.01)	
Alpha Diversity (PD Whole Tree)**	183.8 (6.64)	193.7 (11.33)	
Notes.

* P < 0.10.

** P < 0.05.

*** P < 0.01.

Figure 1 Boxplots are shown for net MeHg production.

Boxplots are shown for net MeHg production (calculated as concentration at 28 days less the initial concentration), with upper and lower hinges representing the values at the 75th and 25th percentiles and whiskers representing 1.5 times value at the 75th and 25th percentiles, respectively. Leachate increased net MeHg production in unvegetated sediment but did not have a large impact within vegetated sediments. Regardless of leachate addition, vegetated sediment experienced an order of magnitude higher rates of net mercury methylation. All samples were spiked with HgCl2. Mean increase in MeHg production in ng per g dry ± standard errors are listed below each box.

Microbiome response to HgCl2 and leachate addition

Over the course of the incubation, microcosms with vegetated, high-C sediment produced over ten times more MeHg than unvegetated sediment microcosms, regardless of leachate amendment (ANOVA P = 0.002, Fig. 1). However, leachate did not stimulate MeHg production in the vegetated environment. Within the oligotrophic, unvegetated environment, mercury methylation was enhanced by leachate within the oligotrophic unvegetated environment with roughly two to four times more production in microcosms receiving leachate as compared to those without leachate.

Community structure changed through time in vegetated and unvegetated environments with leachate (ANOSIM across days 0, 7, 14, 21, 28, veg.: P = 0.001 R = 0.40, unveg.: P = 0.001 R = 0.43, Figs. S2A and S2B), but not those without leachate (veg.: P = 0.02, R = 0.19, unveg.: P > 0.05, Figs. S2A and S2B), indicating no substantial effect from high concentrations of added inorganic mercury on microbiome structure. At day 28, communities in unvegetated microcosms with leachate were different than those without leachate (ANOSIM, P = 0.01, R = 0.54), while microbiome structure in vegetated sediment microcosms only weakly differed between leachate and no leachate groups (P = 0.04, R = 0.22).

Changes in community structure in response to leachate was partially generated by shifts in microbial taxa that are known to contain methylating bacteria. For example, we observed an increase in Clostridia in both environments (Kruskal–Wallis, veg.: FDR-corrected P = 0.003, unveg.: P = 0.018, Fig. 2B, Tables S1–S2) and a decrease in Deltaproteobacteria in unvegetated sediment (Kruskal–Wallis, veg.: FDR-corrected P = 0.36, unveg.: FDR-corrected P = 0.015, Fig. 2A). Clostridia do not typically metabolize sulfate, while the taxon Deltaproteobacteria contains many sulfate-reducing organisms. In particular, Clostridia abundances increased by 3-fold (1.1% to 3.8% of the microbiome) and 10-fold (1.5% to 10.5% of the microbiome), respectively in vegetated and unvegetated environments, driven by increases in nearly all families of Clostridia. SIMPER analysis was confirmative of these changes in the full community—22.9% of 175 SIMPER-identified OTUs belonged to Clostridia (increased from avg. 0.78 OTUs/sample to avg. 17.20 OTUs/sample, Tables S1–S2) while 8% belonged to Deltaproteobacteria (decreased from avg. 8.5 OTUs/sample to 7.4 OTUs/sample, Tables S1–S2).

Figure 2 Boxplots are shown for selected changes in taxonomy in response to leachate addition.

All samples were spiked with HgCl2. Upper and lower hinges represent values at the 75th and 25th percentiles and whiskers represent 1.5 times values at the 75th and 25th percentiles, respectively. Outliers are plotted as points. Shading for each bar denote taxonomy and leachate vs. no leachate. Significant relationships (P < 0.05) are denoted with an asterisk. (A) The addition of leachate decreased the proportion of Deltaproteobacteria and increased the proportion of Clostridia in both vegetated and unvegetated sediment, with greater effects in unvegetated sediment. (B) Within organisms in the ORNL database of putative mercury methylators, we observed changes within the family Peptococcaceae (class Clostridia) in response to leachate addition. Abundance data are present in Table S2.

One family of Clostridia known to contain methylating bacteria (Peptococcaceae), sharply increased with leachate in unvegetated sediment and displayed a similar trend in vegetated sediment (Mann–Whitney U, veg.: uncorrected P = 0.03, unveg.: uncorrected P = 0.03, Fig. 2B, Table S1–S2). These changes were due in part to increases in two closely related methylating OTUs (Mann–Whitney U, Dehalobacter restrictus strain PER-K23, veg.: uncorrected P = 0.04, from an average of 0% to 6% of the reduced set of potential methylating taxa, and Syntrophobotulus glycolicus strain DSM 1351, unveg.: uncorrected P = 0.01, from an average of 0% to 3% of the reduced set of potential methylating taxa) grouped in a single genus by our classification system (Dehalobacter_Syntrophobotulus, Fig. S2). Vegetated sediments also experienced a slight increase in [Clostridium] cellobioparum strain DSM 1351 (uncorrected P = 0.03, from an average of 0% to 1% of the reduced set of potential methylating taxa), while unvegetated sediments displayed an increase in Geobacter bemidjiensis strain Bem (uncorrected P = 0.02, from an average of 24% to 42% of the reduced set of potential methylating taxa) and a decrease in Geobacter sp. M21 (uncorrected P = 0.006, from an average of 15% to 3% of the reduced set of potential methylating taxa). No other OTUs changes in these microcosms.

Metagenomic shotgun sequences were consistent with microbiome shifts observed in 16S rRNA genes. We note increases in Clostridia (t-test, FDR-corrected P = 0.006), Peptococcaceae (t-test, FDR-corrected P = 0.018), Dehalobacter restrictus (t-test, FDR-corrected P = 0.024), and Syntrophobotulus glycolicus (t-test, FDR-corrected P = 0.042) as well as a possible trend for decreases in Deltaproteobacteria (t-test, FDR-corrected P = 0.18) in metagenomic data (Fig. 3D). We also idenitfied 7,150 KEGG pathways, 84 COGs, and 79 Pfams that were significantly enriched at day 28 relative to day 0 in unvegetated leachate microcosms (Figs. 3A–3C). All classfication systems revealed metabolic shifts towards glycosyltranseferases, among other pathways involved in DOM oxidation and in iron and nitrate reduction.

Figure 3 Results from analysis of metagenomic shotgun sequences from unvegetated microcosms are denoted in Fig. 3.

All samples were spiked with HgCl2. (A–C) show the abundance of the top 15 KEGG, COG, and Pfam targets that increased at day 28 vs. day 0, respectively. (D) shows percent change in selected taxonomic groups at day 28 vs. day 0. Error bars denote standard error.

Changes in DOM chemistry

Details of DOM quantity and quality changes are presented in Fig. 4 and Fig. S3 and described in greater detail in Supplemental Information 1. Regression statistics associated with Fig. 4 and Fig. S3 are presented in Table 2.

Figure 4 DOM fluorescence indices were assessed through time with linear and quadratic regressions in each environment and microcosm type.

All samples were spiked with HgCl2. Averages for each environment and microcosm type are plotted at days 0, 7, 14, 21, and 28, with error bars representing the standard error. Plots in the first column are leachate microcosms, while plots in the second column are no leachate microcosms. Unvegetated microcosms are depicted as closed circles with dashed lines showing significant regressions; vegetated microcosms are x’s with solid lines showing significant regressions. (A) and (B) denote FI, (C) and (D) denote HIX, and (E) and (F) denote freshness.

Table 2 Regression statistics (R2 values) from analysis of changes in DOM properties through time are listed in Table 2.

These values are associated with regressions presented in Fig. 4 and Fig. S3. All DOM properties are derived from water overlaying sediments in our incubations. No leachate microcosms were analyzed from across days 7, 14, 21, and 28; and leachate microcosms were analyzed across days 0, 7, 14, 21, and 28 (n = 4–5 at each sampling point, no samples were taken in no leachate microcosms at day zero), with characteristics of the applied leachate represented at day 0.

	NPOC (mg/L)	Total Fluoresence	Fluor:NPOC	FI	HIX	Freshness	
Vegetated, No leachate (across days 7, 14, 21, 28)	0.39**	0.21*	n.s.	0.22**	0.51***	0.52***	
Vegetated, Leachate (across days 0, 7, 14, 21, 28)	0.32***	n.s.	n.s.	n.s.	0.68****	0.57***	
Unvegetated, No leachate (across days 7, 14, 21, 28)	0.64****	n.s.	0.29**	n.s.	n.s.	n.s.	
Unvegetated, Leachate (across days 0, 7, 14, 21, 28)	n.s.	n.s.	n.s.	0.41***	n.s.	0.39***	
Notes.

* P < 0.10.

** P < 0.05.

*** P < 0.01.

**** P < 0.001.

DOM fluorescence indices displayed notable changes through time. In the vegetated environment, FI remained stable at a low value in leachate microcosms, indicating plant-derived DOM, and rose in microcosms without leachate indicating greater relative contribution of microbial vs. abiotic processing (Figs. 4A and 4B). In contrast, in the vegetated environment, HIX increased in both leachate and no leachate microcosms indicating processing of more labile vs. recalcitrant DOM (Figs. 4C and 4D). This increase in HIX corresponded with decrease in freshness index (Figs. 4E and 4F), further supporting our interpretation. In the unvegetated environment, leachate microcosms (but not microcosms without leachate) increased in FI (Figs. 4A and 4B) denoting an increase in microbially-sourced DOM over time. There was no change in HIX (Figs. 4C and 4D) suggesting equal processing of labile vs. recalcitrant DOM. Freshness varied non-linearly in leachate microcosms but not those without leachate (Figs. 4E and 4F).

Across environment types, HIX was significantly higher in vegetated microcosms (ANOVA P < 0.0001, Tukey HSD, leachate: P < 0.0001, no leachate: P = 0.004). FI and freshness were higher in unvegetated leachate microcosms than in vegetated DOM-amended microcosms (Tukey HSD, FI: P = 0.003, freshness: P = 0.03) but did not differ across microcosms without leachate (Tukey HSD, FI: P = 0.89, freshness: P = 0.40).

Correlation of microbiome, DOM characteristics, and MeHg production

Given the apparent shift in community structure towards Clostridia, and chemoorganotrophic Peptococcaceae in particular, we examined correlations of members of this family listed in the ORNL methylator database with the proportion of complex organic matter (HIX) and MeHg production within each environment. We focused on HIX because this index changed consistently and reflected portions of recalcitrant carbon substrate pools utilized by the organisms we identified. Because we only calculated net MeHg production at the conclusion of the incubation, we analyzed these correlations at day 28 and grouped leachate and no-leachate replicates within each environment to provide sufficient variation and sample size (n = 6). Peptococcaceae with the potential to methylate merucry negatively correlated with HIX and positively correlated with net MeHg production in unvegetated microcosms (Pearson’s r (n = 6), HIX: r =  − 0.82, MeHg: r = 0.67). The same organisms were not strongly correlated with HIX (r =  − 0.49) or net MeHg production (r =  − 0.04) in vegetated microcosms.

Table 3 Relationships of selected COG and Pfam targets with HIX and net MeHg production at day 28 (n= 3) are presented in Table 3.

Values are Pearson’s r, and significance levels are denoted by asterisks.

	HIX	MeHg	
COG			
Glycosyltransferase	−0.98*	0.79	
Glycosyltransferases involved in cell wall biogenesis	−0.96*	0.76	
ABC-type nitrate/sulfonate/bicarbonate transport systems. periplasmic components	−0.85	0.55	
FOG: PAS/PAC domain	−0.88	0.60	
Predicted metal-dependent hydrolase of the TIM-barrel fold	0.88	0.60	
Transcriptional regulator	−0.999**	0.88	
Outer membrane receptor proteins. mostly Fe transport	−0.79	0.45	
HD-GYP domain	−0.99**	0.84	
Glycosyltransferases. probably involved in cell wall biogenesis	−0.96*	0.74	
Beta-galactosidase/beta-glucuronidase	0.98*	−0.80	
Sugar phosphate isomerases/epimerases	−0.73	0.36	
Lactoylglutathione lyase and related lyases	−0.93	0.67	
Nitroreductase	−0.996**	0.86	
Thiamine biosynthesis enzyme ThiH and related uncharacterized enzymes	0.98*	0.80	
ABC-type phosphate transport system. periplasmic component	−0.66	0.27	
Pfam			
WD40-like Beta Propeller Repeat	−0.99**	0.85	
Glycosyl transferase family 2	−0.97*	0.76	
TonB dependent receptor	0.90	−0.9999***	
Radical SAM superfamily	−0.95*	0.75	
TonB-dependent Receptor Plug Domain	0.51	−0.83	
Amidohydrolase	−0.87	0.57	
NMT1/THI5 like	−0.87	0.58	
HD domain	−0.9997**	0.91	
DNA gyrase C-terminal domain. beta-propeller	−0.94	0.70	
Protein of unknown function (DUF1501)	−0.46	0.04	
RHS Repeat	−0.80	0.46	
Doubled CXXCH motif (Paired_CXXCH_1)	−0.97*	0.78	
Helix-turn-helix	−0.90	0.63	
Natural resistance-associated macrophage protein	−0.83	0.51	
SusD family	0.99*	−0.82	
Notes.

* P < 0.10.

** P < 0.05.

Finally, despite low statistical power (n = 3), we observed marginally significant trends (P < 0.10) between key metabolic pathways and HIX (Table 3). While we note that the sample size for this analysis was low, it is remarkable to observe any trends under this limitation and we provide results as an encouraging avenue for future research. In particular, COGs classified as Glycosyltransferase, Glycosyltransferases involved in cell wall biogenesis, Glycosyltransferases—probably involved in cell wall biogenesis, and Beta-galactosidase/beta-glucuronidase; and Pfams classified as Glycosyl transferase family 2, Radical SAM superfamily, and SusD family displayed significant correlations with HIX at the P < 0.10 level. Only Pfam PF00593, TonB dependent receptor, correlated with MeHg production (P < 0.001, r =  − 1.00, Table 2).

Discussion

We show the importance of vegetation patterns and DOM availability in mercury cycling within Lake Superior’s St. Louis River Estuary, an integral environment to human society and industries of the region. Our work demonstrates not only the far higher levels of mercury cycling in natural vegetated over unvegetated sediments, but also the susceptibility of oligotrophic, unvegetated sediments to increases in mercury methylation and changes in microbiomes with the addition of DOM. We also suggest a possible involvement of metabolisms that ferment recalcitrant organic matter (OM) in mercury methylation, particularly within oligotrophic unvegetated environments. Our results provide a basis for further investigation into the role of newly discovered microorganisms in regulating the production of MeHg in the Great Lakes region and further a body of work aimed at understanding and mitigating human exposure to MeHg.

Mercury methylation across environments

Our work indicated a strongly different capacity of vegetated vs. unvegetated wetland sediments to cycle mercury. Without leachate addition, MeHg production in vegetated sediments was two orders of magnitude higher than in unvegetated sediments (Fig. 1). As such, vegetated sediments may be considered potentially important locations for mercury methylation in contaminated watersheds. Such a dynamic may be due to either higher overall activity of microorganisms or the unique microbiomes contained within these sediments. Within the high-C vegetated environment, leachate did not influence the sediment microbiome or net MeHg production to the same extent as within the more oligotrophic unvegetated environment (Fig. 1, Fig. S1). Given high ratios of C:N, high OC content, and low NO3− concentrations in our vegetated sediment (Table 1), N-limitation may have mitigated net MeHg production in vegetated environments relative to the unvegetated environment (Taylor & Townsend, 2010), which had substantially lower concentrations of all measured C and nutrient concentrations. Both ambient MeHg concentrations prior to microcosm amendment and net MeHg production were dramatically higher in the vegetated environment, supporting other findings that plant-microbe interactions facilitate MeHg production (Gentès et al., 2017; Roy, Amyot & Carignan, 2009; Windham-Myers et al., 2014; Windham-Myers et al., 2009).

By contrast, the unvegetated environment experienced a dramatic increase in MeHg (Fig. 1) in response to leachate that correlated with changes in the sediment microbiome (Figs. 2 and 3, Fig. S1 ). Carbon limitation has been widely demonstrated as a constraint on microbial activity (Bradley, Fernandez Jr & Chapelle, 1992; Brooks, McKnight & Elder, 2005; Wett & Rauch, 2003); thus, leachate may bolster MeHg production in C-limited ecosystems via impacts on microbial activity. In our system, net MeHg production in the unvegetated environment was possibly also constrained by low in situ rates of microbial activity and by low N concentration, and net MeHg production in response to leachate stimulus never increased to vegetated levels. Importantly, leachate enhanced the relative abundance of a specific taxon known to contain methylating organisms (Clostridia), raising the possibility that mercury methylation rates may be dually influenced by the sediment microbiome and by OM (Aiken, Hsu-Kim & Ryan, 2011; Hsu-Kim et al., 2013).

Microbiome response to leachate addition.

Microbiome responses to leachate in both sediment types are consistent with recent work demonstrating the fermentation of OM by Clostridia despite the presence of OM-oxidizing Deltaproteobacteria (Reimers et al., 2013) and suggest a possible role for members of Clostridia in MeHg production, either through direct methylation or indirectly by enhancing the availability of OM for other organisms through fermentation. Within both environments, leachate altered the sediment microbiome, driven largely by increases in Clostridia and decreases in Deltaproteobacteria. Unvegetated microcosms displayed greater changes in these clades, supporting a greater role for environmental filtering by DOM within oligotrophic environments (Barberán et al., 2012; Graham et al., 2016a; Graham et al., 2017; Stegen et al., 2012). Clostridia are obligate anaerobes with the ability to produce labile carbon compounds via fermentation of recalcitrant OM (Reimers et al., 2013; Ueno et al., 2016). Recent work has shown organic carbon degradation via Clostridial fermentation to operate at comparable rates to more energetically favorable carbon processing pathways (Reimers et al., 2013). Organic acids (e.g., lactate and acetate) produced through these pathways can also be utilized as a carbon source by sulfate- and iron- reducing Deltaproteobacteria (Guerrero-Barajas, Garibay-Orijel & Rosas-Rocha, 2011; Reimers et al., 2013; Zhao, Ren & Wang, 2008). Thus, enhanced DOM breakdown by Clostridia may support other biogeochemical processes (e.g., sulfur, iron, and nitrogen cycles) that rely on organic carbon as an energy source.

In unvegetated sediments, metagenomic analyses indicated an increase in carbon, and secondarily, iron metabolisms consistent with clades known to methylate mercury, although no methylating pathways could be identified in this work (Gilmour et al., 2013; Hamelin et al., 2011; Kerin et al., 2006; Podar et al., 2015). Carbon metabolisms were the primary KEGG category increasing in abundance within metagenomes (Fig. 3A), and several COG pathways and Pfams indicated a possible metabolic shift favoring glycosyltransferases that convert starches, sugars, and nitroaromatics into a wide range of compounds (Bowles et al., 2005; Ramli et al., 2015) (Figs. 3B and 3C). Further, metagenomic increases in Beta-galactosidase/beta-glucoronidase (lactose to galactose/glucose) (Martini et al., 1987), sugar phosphate isomerase/epimerases (sugar metabolism) (Yeom, Kim & Oh, 2013), and lactoylglutathione lyase (detoxification for methyglyoxal fermentation byproduct) (Inoue & Kimura, 1995) and the SusD family (glycan binding) (Martens et al., 2009) provide additional evidence for increases in fermentation processes in response to leachate. Increases in TonB dependent receptors (Moeck & Coulton, 1998), amidohydrolase (Seibert & Raushel, 2005), and NRAMP (Cellier et al., 1995) suggest a secondary importance of iron processing and/or transport of large organic compounds across cellular membranes. Finally, our results provide a possible genetic mechanism connecting iron, sulfur, carbon, and mercury cycling, as the radical SAM superfamily, which facilitates methyl transfers via the use of a [4Fe-S]+ cluster (Booker & Grove, 2010), increased in concert with net MeHg production. In total, the metabolic potential of the sediment microbiome indicates changes in carbon and iron metabolisms within microcosms experiencing higher net MeHg production in response to leachate, supporting past work that suggests a linkage between mercury methylation and these factors (Gilmour et al., 2013; Hamelin et al., 2011; Hsu-Kim et al., 2013; Kerin et al., 2006; Podar et al., 2015).

Lastly, at high taxonomic resolution in both environments, leachate increased the proportion of bacterial taxa that are known to contain methylating organisms such as Peptococcaceae within Clostridia, despite drastic differences in sediment chemistry (Fig. 2B). Specifically, the two OTUs identified by ORNL as organisms with mercury methylation genes that displayed the greatest change are thought to generate energy via organohalide respiration (D. restrictus) and fermentative oxidation of organic matter (S. glycolicus, also capable of syntrophy) (Han et al., 2011; Stackebrandt, 2014). The relative abundance of Peptococcaceae was positively correlated with net MeHg production in the unvegetated environment, and other taxa that are known to contain methylating organisms did not increase in abundance, as would be expected if the activity of these organisms was enhanced by leachate.

We note that shifts in these taxa contain many OTUs that are not methylating bacteria, however, we attempt to focus our analysis of changes in microbiome composition to taxa that are relevant to methylation. Overall this work points to the effects of DOM on microbial community composition with potential implication for microbiome function that may influence mercury cycling.

Associations between microbiology, DOM processing, and net MeHg production

The processing of proportionally more labile (microbe-preferred) OM would be expected to result in increases in HIX. However, changes in these indices within the unvegetated environment suggest substantial recalcitrant OM degradation vs. the metabolism of labile substrates (but not in the vegetated environment which followed the expectation of increase HIX). We observed no change in HIX through time in unvegetated microcosms (both leachate and no leachate). This result is reflective of a DOM pool that has stable relative proportions of labile and recalcitrant OM, indicating equal rates of degradation and/or production of both substrate types (Figs. 4C and 4D). Vegetated microcosms, in contrast, experienced increases in HIX through time that indicate a loss of labile substrate from the fluorescent DOM pool (Figs. 4C and 4D). Further, in leachate unvegetated microcosms, which experienced pronounced changes in the sediment microbiome and increased MeHg production, HIX was significantly lower than in all other experimental groups (ANOVA, P < 0.0001, all Tukey HSD P < 0.0001). While most microorganisms preferentially degrade labile C sources, the degradation of recalcitrant OM can contribute substantially to aquatic carbon cycling (Mcleod et al., 2011). Unvegetated microcosms receiving leachate also exhibited large increases in microbially-derived DOM (FI) through time, demonstrating a noticeable contribution of microbial activity to the DOM pool (Fig. 4A).

The abundance of Peptococcaceae in unvegetated microcosms negatively correlated with HIX, denoting an apparent association of these members or co-occuring community members with DOM processing, but the mechanisms behind these shifts remain unclear. Metabolism of recalcitrant OM by fermenting organisms may influence mercury methylation via direct and indirect mechanisms. Members of Clostridia can generate MeHg themselves, and Clostridial degradation of recalcitrant OM can also produce bioavailable carbon substrates for sulfate- and iron- reducing organisms that produce MeHg.

While further work with a larger sample size is needed, changes in metagenomes in response to leachate denote interesting metabolic pathways that may be involved in recalcitrant OM processing and MeHg production. For example, both COG and Pfam glycosyltransferases were negatively correlated with HIX, suggesting a role for starch, sugar, and nitroaromatic fermentation in response to DOM loading. As well, a negative correlation between HIX, and the radical SAM superfamily provides a possible mechanistic linkage between methyl transfers and recalcitrant organic matter processing. Conversely, Beta-galactosidase/beta-glucuronidase, and the SusD family were positively correlated with HIX, indicating a co-association with labile C processing rather than recalcitrant OM.

Conclusions

Our work shows clearly distinct mercury cycling dynamics between the vegetated and unvegetated sediments of the St. Louis River Estuary. While substantially greater MeHg production is observed in vegetated sediments, unvegetated sediments stand to respond more strongly to DOM additions in driving increases in MeHg production. We also describe changes in DOM pool properties through time using fluorescence indices that can be readily applied in natural systems and may be particularly valuable for monitoring efforts in wetlands of the Great Lakes Region. Moreover, we observed changes in the microbiome of both high-C and oligotrophic sediment in response to leachate addition. The oligotrophic environment showed greater responses in the sediment microbiome and in mercury methylation to the addition of DOM, an important insight given increasing risks of anthropogenic eutrophication (Hsu-Kim et al., 2018; Obrist et al., 2018). Microbiome shifts towards fermentation pathways, increases in chemoorganotrophic Clostridia, degradation of recalcitrant OM, and increases in MeHg within oligotrophic environments emphasizes the need to further study microbial ecology of mercury methylation. While a correspondence between Clostridia, fermentation metabolisms, and MeHg does not necessitate a direct relationship between these processes, the abilities of some Clostridia to methylate mercury and of fermentation products to facilitate other metabolic pathways commonly associated with mercury methylation merits future investigation into the ecology of these organisms. Importantly, our results provide evidence that microbial abundances that correspond with increased mercury methylation include taxa that are known to contain methylating bacteria but are not historically considered in MeHg production. Taken together, our research provides new insights on how DOM may influence microbiome structure and activity differently in two sediment types, impacting MeHg production in natural settings in the Great Lakes region.

Supplemental Information

Supplemental Information 1 Supplemental Info

Click here for additional data file.

We thank Alan Townsend, Teresa Bilinski, Deb Repert, Dick Smith, Steve Schmidt, Sharon Collinge, Garrett Rue, Jess Ebert, Alexis Templeton, and the LSNERR staff for valuable support and feedback during this project. We also thank Axios Review Service for valuable feedback on this manuscript.

Additional Information and Declarations

Competing Interests

Author Contributions

Data Availability

The authors declare there are no competing interests.

Emily B Graham conceived and designed the experiments, performed the experiments, analyzed the data, contributed reagents/materials/analysis tools, prepared figures and/or tables, authored or reviewed drafts of the paper, approved the final draft.

Rachel S. Gabor analyzed the data, contributed reagents/materials/analysis tools, authored or reviewed drafts of the paper, approved the final draft, knowledge on fluorescence and code development.

Shon Schooler conceived and designed the experiments, contributed reagents/materials/analysis tools, authored or reviewed drafts of the paper, approved the final draft, local knowledge and field support.

Diane M. McKnight and Diana R. Nemergut conceived and designed the experiments, contributed reagents/materials/analysis tools, authored or reviewed drafts of the paper, approved the final draft.

Joseph E. Knelman conceived and designed the experiments, performed the experiments, contributed reagents/materials/analysis tools, authored or reviewed drafts of the paper, approved the final draft.

The following information was supplied regarding data availability:

Raw sequences and mapping file are available at figshare: https://figshare.com/articles/Sequences/5833935.

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
