# Peer review of "Oligotrophic wetland sediments susceptible to shifts in microbiomes and mercury cycling with dissolved organic matter addition"

_PeerJ, doi:10.7717/peerj.4575_

## Round 0.1 · original submission · Minor Revisions

This is a very nice manuscript and the suggested revisions are rather minor. I look forward to seeing your revised manuscript. Please let me know if you need any clarification on the reviewer comments.

·

Basic reporting

This is a nicely written clear account of a factoral experiment to determine the impact of plant leachates on methylmercury production in a freshwater wetland.

There are a few newer reviews that should be included in subsequent revisions.

Symbols on figures would be better a bit larger.

Experimental design

The subject is entirely appropriate for PeerJ.

The methods need some work. While many of the methods included in the supplimental materials should stay there, there is quite a bit of details required in the main paper to make the methods clear. I have added what I think is needed in comments on the text file.

Validity of the findings

I can not comment on the statistical analysis.

The data is robust.

I think that there needs to be some work on the implications of the use of 16S methods to determine the microbes present vs. the fact that just because they are present, doesn't mean they actually methylate. More details in comments in the text file.

Additional comments

Nice paper. I look forward to seeing the revised manuscript if deemed appropriate by the editors.

Reviewer 2 ·

Basic reporting

This is a well-written manuscripts on the impacts of DOM to mercury methylation in vegetated and unvegetated microcosms.

Experimental design

Experimental design was well done. Statistical tests were generally correct and sound.

Validity of the findings

The findings on the effects of leachate on microbial community structure is the most important finding different from previous study.

Additional comments

Line 53: citation by publication time or alphabetic order of first author?
Line 67-68: need to know where is St Louis River, there is another one Florida.
Line 188 and Line 191: “at XXXXXX”?
Line 227: why was SUVA>7 removed?
Line 249-250: you may use Spearman Rank correlation if data were not normally distributed.
Line 271: should be ug/g wt sediment.
Line 279-280: leachate did not stimulate MeHg in vegetated sediment but did increase MeHg in oligotrophic unvegetated environment. Why?
Line 292-303 This is a multi-discipline study. Maybe not all the readers understand all aspects of the technical work. I would like to see for example, which group of methylators require sulfate and not require sulfate. Giving the name of the microbial group is not sufficient.
Line 388: it seems this is contradictory to Line 279-280 if I don’t read it wrong.
Fig 1: +/- may be expressed as ±

---

## Round 0.2 · accepted · Accept

It was a pleasure serving as your editor. Quality manuscripts such as yours make my job easy.

#